# GAN-BASED NERF NOISE SIMULATION IN MESH DE-NOISING TASK

## ABSTRACT

In the present paper, we propose a new approach and a dataset for generating NeRF-like noise on the mesh surface. Our approach is based on GAN and was trained on a dataset that we collect using real NeRF noise. The core idea of our method lies in the use of graph convolutions in the generator. Our pipeline demonstrates generated NeRF-like noise more accurate than other methods by mesh denoising benchmarking. We also present a new NeRF noise analysis approach HTPH based on a conditional probability model to measure the similarity of mesh noise.

## 1 INTRODUCTION

The problem considered in this article belongs to mesh denoising and 3D scene reconstruction domains. Existing scene reconstruction algorithms work with errors called noise Nguyen et al. (2012); Belhaoua et al. (2009). In our work, we focus on the node noise which causes the difference between shapes of real and reconstructed objects.

Neural radiance fields (NeRF) models have recently become a popular tool for reconstructing 3D scenes. The first one was introduced in Mildenhall et al. (2021). Many NeRF extensions were presented in the next several years after the first publication Müller et al. (2022); Barron et al. (2021); Jain et al. (2021); Deng et al. (2022); Yu et al. (2021); Fridovich-Keil et al. (2022); Wang et al. (2022); Zhi et al. (2021); Jeong et al. (2021). In our paper we consider NeRF as a baseline algorithm for 3D scene reconstruction by a set of views.

The experiments show that 3D object reconstruction using NeRF requires a lot of time. There are NeRF models where learning a scene takes up to 40 hours. Therefore, a lot of time is required to generate enough data to train the noise reduction algorithm specified for NeRF noise.

Our motivation is to reduce the time-consuming process of data generation. For this purpose we introduce a pipeline that can generate a dataset in a short time. Using our approach we generate a new dataset. We have shown that mesh denoising models trained on our dataset remove a noise appearing in the scene after NeRF reconstruction better than without training on our dataset. Apart from this, we present a new mesh noise description based on conditional probability model which we have used in our analysis.

Our pipeline is based on Generative Adversarial Network (GAN) Goodfellow et al. (2014). To train our model we select meshes from objaverse-XL Deitke et al. (2023) and preprocessed them. We use Instant-NGP Müller et al. (2022) to prepare NeRF noise examples.

Our **main contribution** can be summarized as follows:

- We introduce a **new analysis of mesh noise** which uncovers the significant difference between artificial noise and real noise.

- We propose a new pipeline for generation NeRF-like noise on the mesh surface. The core of our pipeline is a GAN which was trained on real NeRF noise. The **application** consists in the massive generation of a dataset suitable for effective training of denoising models.

## 2 RELATED WORK

In recent years, learning-based mesh denoising methods have achieved impressive results, particularly: DNF-Net Li et al. (2020), NormalNet Zhao et al. (2021), IMD-Net Botsch et al. (2022), GeoBi-GNN Zhang et al. (2022), Cascaded Regression Wang et al. (2016) and GCN-denoiser Shen et al. (2022). All learning-based methods require a large number of clean-noisy mesh pairs, which are complicated and time-consuming to acquire. Mesh noise generation methods do not require as much time as in-camera processing pipelines.

Existing noise generation methods can be divided into the following groups:

### 2.1 NON-LEARNING-BASED NOISE MODELS

The literature review shows that a probability density function (PDF) is typically used for non-learning-based modeling of sensor noise. The PDF parameters are determined through experimental measurements.

A Konica Minolta Vivid 910 3D laser scanner is considered by Sun et al. (2008). The authors plot the histogram of noise magnitudes and interpolate the PDF using Gaussian distribution. The Microsoft Kinect noise is analysed by Nguyen et al. (2012). The authors demonstrate how distribution depends on the angle of rotation and the distance between the sensor and the plane. Choo et al. (2014) creates another noise model of a Microsoft Kinect depth sensor. The authors use the chessboard for experiments and show how noise distribution depends on the depth of scene points. Haider & Hel-Or (2022) create a noise histogram from a series of measurements from different sensor positions and show that the noise distribution depends on the light direction and distance. The authors compare the noise distributions of three types of depth sensors: ZED, Microsoft Kinect V1 and Microsoft Kinect V2.

### 2.2 LEARNING-BASED NOISE MODELS

The noise can be learned directly with GAN if the noise is too complicated and cannot be modeled as PDF. In particular, noise generation is often applied to images when white noise generation is required.

Henz et al. (2020) construct the GAN model, in which a generator consists of five sequential residual blocks, two convolutional and one batch normalization layer. Each residual block contains two convolutional, two normalization layers and a ReLU layer. At the same time, the discriminator consists of five convolutional layers, each followed by an instance normalization and a leaky ReLU layer. Similarly, Tran et al. (2020) uses the same model with five residual blocks for image noise generation. Kim et al. (2019) introduces a generator with sequential residual blocks and convolutional blocks where each convolutional block has batch normalization, spectral normalization and ReLU layers. At the same time, each residual block has two $3 \times 3$ convolutional layers.

In contrast, some researchers use U-Net-based (Ronneberger et al. (2015)) model as a generator. Hossain & Lee (2022) creates a U-Net-based model with 10 blocks, where each block contains a channel attention layer, two recurrent convolutional blocks with batch normalization, and ReLU layers. Chang et al. (2020) uses a camera-encoding network in addition to U-Net-shaped generator for realistic camera noise generation. Song et al. (2023) builds U-Net for camera noise generation with six SNAF blocks. Each block has three convolutional, one normalization, and one simple gate layers. The Decoder blocks have additional noise injection layers.

## 3 MESH NOISE ANALYSIS

In this section, we present an approach to compare mesh noise that takes into consideration the dependence between neighboring vertex positions. Our approach uncovers the significant difference between artificial mesh noise and realistic noise caused by the weaknesses of algorithms and sensors.

We refer to the vertex offset as the distance between original mesh and noised mesh for each vertex in the noisy mesh. The method for calculating the distance between point and mesh is described in the Appendix 1. This approach is based on Zong et al. (2023). Most existing denoising works

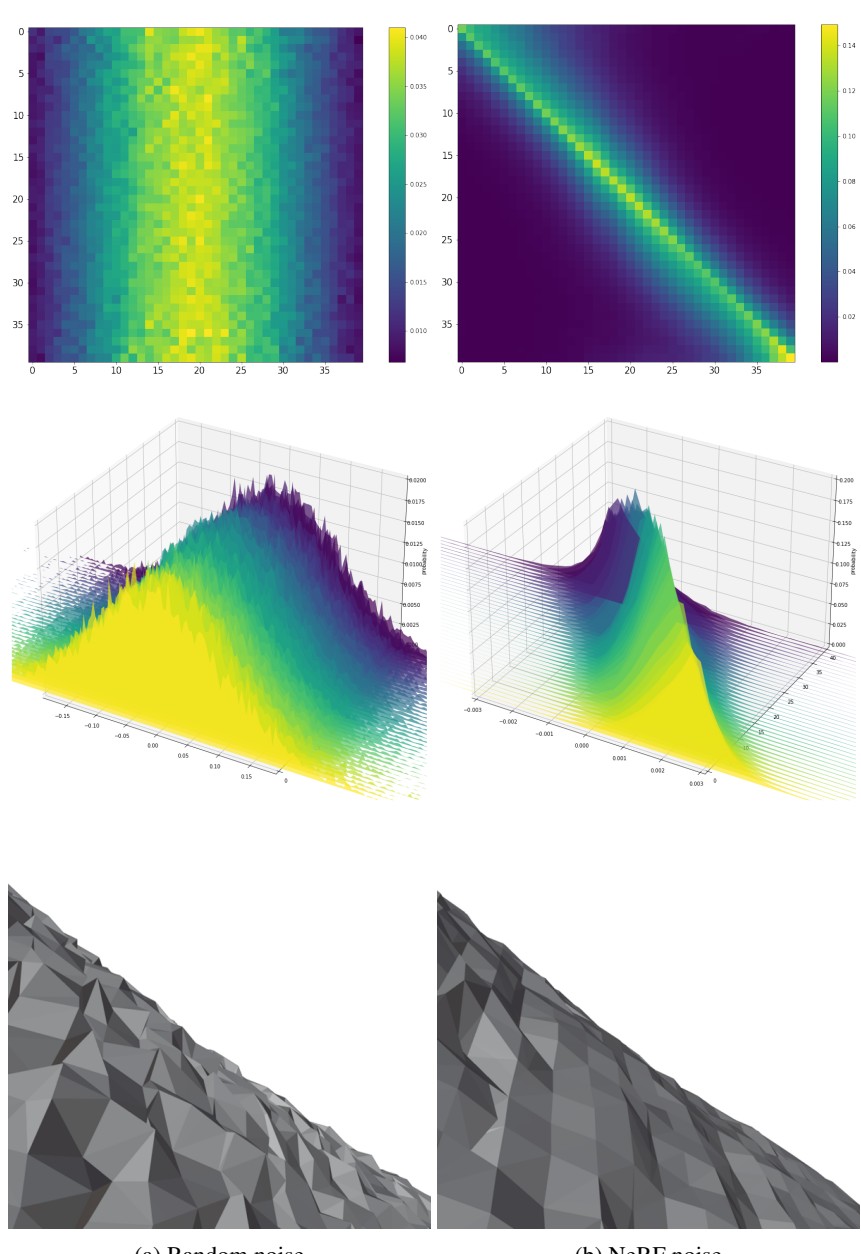

(a) Random noise                    (b) NeRF noise

Figure 1: Random normal noise heatmap (a) and NeRF noise heatmap (b). Each heatmap is a transition probability matrix, where the vertical axis defines a vertex offset class $C(v)$, and the horizontal axis defines offset classes of neighboring vertices. Thus, each cell defines the probability of two different offsets to be neighboring. For all classes, the distribution is normal with approximately the same expected values and standard deviations. In contrast, the MTPH of NeRF noise shows that neighboring vertices always have sufficiently close offsets.

have noise generation tools for dataset generation. Unfortunately, the algorithms in these tools only the offset of individual vertices without taking into account the offset of neighboring vertices. Thus, these algorithms are only able to imitate unrealistic noise modeled by single vertex offsets distribution, and the dependence between offsets of neighboring vertices is not taken into account. A simple offsets distribution does not show the difference between artificial mesh noise and realistic noise. We have developed a new approach to measure the difference between types of noise.

We denote by $s(v)$ the offset of the vertex $v$. All vertex offsets can be arranged in ascending order, and we can find $s_{min}$ and $s_{max}$ along all vertices. The segment $[s_{min}, s_{max}]$ is divided into $K$ equal sub-segments. Each sub-segment represents a class of vertices to which offsets belong. Therefore, we divide all vertices into $K$ different classes determined by natural numbers. We define a class index for each vertex as $C(v) = \left\lfloor (s(v) - s_{min})/d \right\rfloor$, where $d = (s_{max} - s_{min})/K$, i.e. $C \in \{0, \ldots, K-1\}$. In practice, $s_{min}$ is calculated as the 4th percentile and $s_{max}$ as the 96th percentile so that each class does not contain too few vertices.

We denote all neighboring vertices for each vertex $v$ as $N(v)$. We define a dependence between the offsets of neighboring vertices as a conditional probability model. We consider the vertex classes $C$ as states of the model. Let's define state transition probabilities: for each vertex $v$ we calculate the class $C(v)$ and the classes $C(v')$ for $v' \in N(v)$. Considering all vertices, for each vertex class we construct the distribution of classes of neighboring vertices. Therefore, we define a state transition probability distribution for each state.

The state transition probability distributions can be represented as a transition probability matrix, where each element $p_{ij}$ indicates the probability of transition from state $i$ to state $j$. We represent this matrix as a heatmap and call it Mesh Transition Probability Heatmap (MTPH).

The MTPH shows the difference between NeRF noise and random noise artificially generated for each vertex, without dependence on neighboring vertices. We calculate the MTPH for meshes from synthetic datasets and NeRF datasets collected by our program. The results for $K = 40$ are shown in Figure 1. It can be seen that the transition probability distribution in artificially generated noise does not depend on the vertex class.

The difference between noise can be measured by the distance between MTPHs. We use the following metrics: cosine difference, Euclidean distance, Manhattan distance, and kernel norm of MTPH difference. Along with MTPH metrics, we use KL divergence to measure the distance between the vertex offset distributions. Five metrics in total. We use these metrics to measure the distance between generated noise and real noise.

# 4 FULL PIPELINE

The task of noise simulation requires a generative model to capture intrinsic noise features during the training process. We choose GAN because there are already published works such as Song et al. (2023) where authors used GAN to generate digital camera noise synthesis on images. It makes sense to refer to the experience of the neighboring domain, which is why we recommend using GAN specifically for noise synthesis.

In our approach we use GAN's generator to predict offsets for point clouds. We transform the mesh into a point cloud which is given to the GAN input. After GAN calculates offsets, we transform noisy point clouds back to mesh. The **scheme of our pipeline** is depicted on Figure 2.

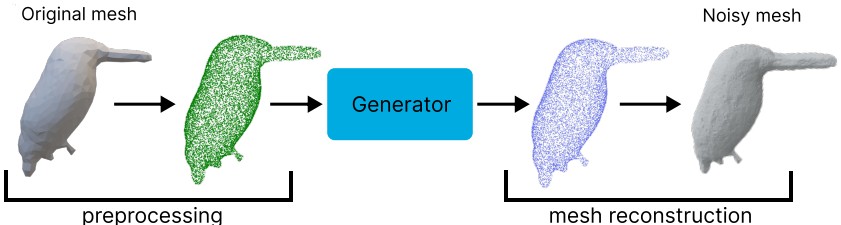

Figure 2: The mesh noise generation pipeline scheme. In the start green point cloud is produced by original mesh point sampling. The Generator takes the green point cloud, calculates the offset for each point, and returns the blue point cloud. The number of points in green and blue clouds is equal. Finally, the noised mesh is reconstructed from the blue point cloud.

### 4.1 GAN ARCHITECTURE

In our method, the generator predicts the magnitude of the offset for each point along the point's normal, so to get a noisy point cloud we need to multiply these magnitudes with point normals and add the resulting offsets to original point positions. The first step to generate a noisy point cloud is to encode the original point cloud using two PointNet layers, each of those utilize the point cloud and its knn-graph. Then, to the encoded point cloud we add random values sampled from random distribution as suggested in study Song et al. (2023). The noisy point cloud features are then fed to GATv2 Layer Brody et al. (2022) and head layers in the end.

The discriminator requires point offsets and the original point cloud knn-graph as input. First, it propagates through the knn-graph using offsets as node features, and then follows a single linear plane and global mean pooling operation to obtain a vector representation of each point cloud in a stack. The point cloud vector representation is then fed to linear layers, followed by sigmoid activation to predict the probability of generating a point cloud. The schematic of the generator and discriminator is shown in the Figure 3.

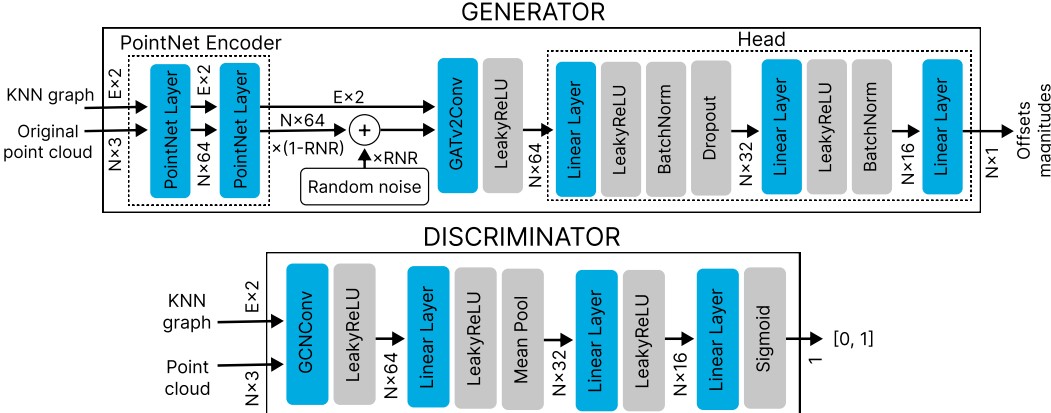

Figure 3: Architecture of GAN: E denotes the number of knn-graph edges, N denotes the number of points, RNR means Random Noise Ratio, which defines the quantity of injecting noise.

## 5 GAN TRAINING

In this section we describe how we prepare clear/noisy pairs to train GAN. All steps are shown on the Figure 4. We also highlight in this section the training details.

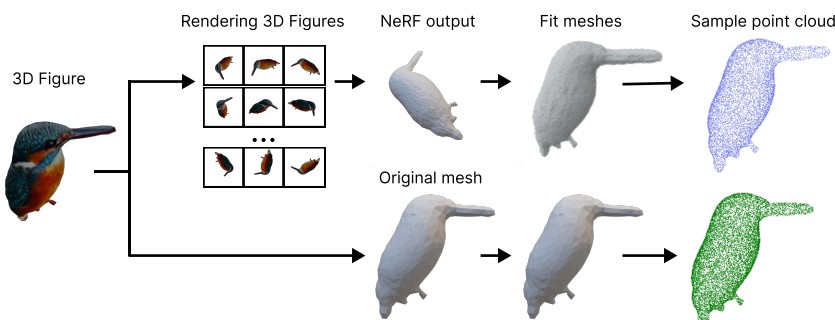

Figure 4: All steps of clear/noisy pairs preparation to train GAN. A mesh is rendered from 100 points of view. The renders are given to NeRF. The reconstructed mesh is fitted to the original mesh. Further, each mesh is used to sample point clouds.

## 5.1 Clear/noisy pairs preparing

We select meshes from objaverse-XL that satisfy the following conditions: they must be watertight and textured, have Euler-Poincar'e characteristic not greater than 10, and must have not less than 500 and not greater than 250 000 vertices. Each mesh is rendered 100 times from different viewpoints using our renderer based on the bpy library in Blender (2018).

The rendered images are used as input for the NeRF. As NeRF produce a radiance field which is a raw data for 3D-reconstruction algorithms we extract meshes with Differentiable Poisson Surface Reconstruction (DPSR) Peng et al. (2021) (which calculates Signed Distance Function (SDF) Slavcheva et al. (2016)) and Marching Cubes (MC) Lorensen & Cline (1998).

Further we fit mesh pairs using a rigid point-set registration approach (Myronenko & Song (2010)). Let us define a transformation function $T(p, v) = SAv - b$ for each pair that transforms the original mesh vertices, which depends on parameters $p$ defined as follows: $b = (b_1, b_2, b_3)$ is a displacement vector, $A = A(\psi, \theta, \phi)$ is a rotation matrix defined by three independent parameters (Euler angles) and $S = diag(s_1, s_2, s_3)$ is a diagonal matrix defined by three scale factors. The parameters $p$ are found as the $argmin$ of the functional:

$$F(p) = \frac{1}{|V|} \sum_{v \in V} \min_{v' \in V'} ||T(p, v) - v'||,$$

where $V$ and $V'$ are sets of vertices of original mesh and noised mesh respectively. The result of optimal transformation application is depicted in Figure 5a.

To create a pair of noisy and clear point clouds, we uniformly sample 10 000 points on clear mesh. Then we project these points onto the noisy mesh along each point normal, which is calculated as the interpolation of facets vertices normals, for a facet that contains the corresponding point. You can see the visualization of this process in Figure 5b. Thus, we obtain two point clouds of the same size and we can build a bijection from points sampled on original mesh and ones sampled on noisy mesh.

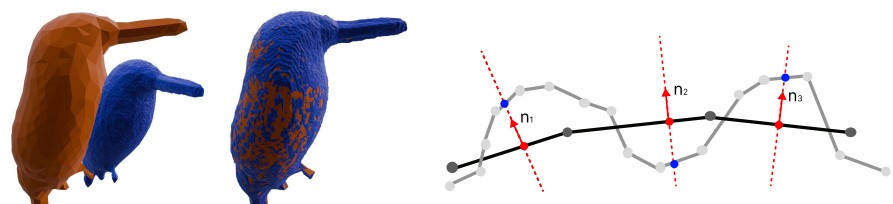

(a) Pair of meshes before alignment and after alignment.

(b) Points sampling process. Original mesh is dark gray and the noisy mesh is light gray.

Figure 5: The rigid point-set registration performance and points sampling illustration.

## 5.2 Training details

We have trained the GAN minimizing the binary cross entropy (BCE) loss and the maximum of the offsets magnitudes. The second term in the loss is needed to deal with outliers. We use the batch size of 16 and the learning rate of $2 \times 10^{-4}$. The experiments conducted on NVIDIA A100 80 GB GPUs, the training process with such settings requires approximately 8 gigabytes of GPU memory, training takes about 30 minutes on a dataset with 856 objects.

We use the Adam optimization algorithm with $\beta_1 = 0.5$ and $\beta_2 = 0.999$. We use LambdaLR with $\lambda = 0.986^{epoch}$ to schedule a learning rate. We conduct the hyperparameter search via Optuna Akiba et al. (2019), we investigate random noise ratio, dropout probability and latent dimensionality.

The training scheme is depicted on Figure 6.

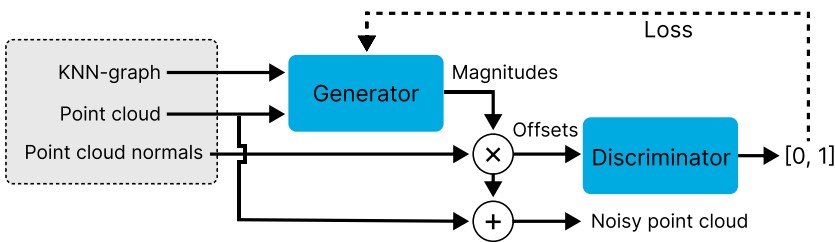

Figure 6: Scheme of the training and inference pipeline. The gray area represents the result of the data preparation pipeline.

## 6 EXPERIMENTS

We conduct a series of experiments to verify that our pipeline can produce a realistic NeRF-like noise.

We select 6 shapes from our primary set for the training and testing domains: bird, bottle, key, sphere, doll and spiral. More details about these shapes can be found in Appendix 2. Each shape was rendered and processed with instant-ngp 1000 times. Each shape was reconstructed after instant-ngp application. We split the shapes to training and testing domains: the bird, bottle, key and sphere are in the train domain and doll and spiral are in the test domain.

We collect offsets in the interval $[-0.004, 0.004]$ to compare the noise distribution histograms. We have defined experimentally that most of the offsets belong to this interval. The code and all clear/noisy pairs are available in our repository[1].

### 6.1 MAIN RESULT

Five types of GAN have been trained: four on one shape – bird, bottle, key, sphere and one on all four train shapes together. Each generator was tested on five train shapes and two test shapes. There are three types of tests we perform:

- In domain (ID) – testing only on the shape that was used for train;
- Out of domain (OOD) – testing on all train shapes except the one that was used for train;
- Test domain (TD) – testing on testing shapes;

The target distribution and target MTPH that we measure distance to are always calculated for the shapes that we test on. The GAN training results are shown in Appendix 3.

### 6.2 ABLATION STUDY

Our main goal is to build a learning-based pipeline that outperforms the baseline method based on DPSR + MC. Furthermore, we compare our GAN results with simple noise generators. The first one is a KNN-regressor, where we choose the number of neighbors of 10. The second one is a simple multilayer perceptron network with four layers (3 to 32, 32 to 16, 16 to 8, 8 to 1), Leaky ReLU activations and batch normalization.

Another baseline for comparison is U-Net, which was also trained in a supervised manner. We reimplement the original architecture presented in Ronneberger et al. (2015) to process 1-dimensional point cloud data instead of 2-dimensional pictures. The baseline is as follows. First, we process raw point cloud with its normals and the knn-graph through PointNet-like encoder, then we feed points' embeddings into reimplemented U-Net, finally the processed embeddings are transformed through two fully connected layers.

You can see the performance of these pipelines in Appendix 3.

---

[1] https://anonymous.4open.science

## 6.3 DISCUSSION

We use our new analysis approach based on MTPH to compare the performance of pipelines. The Table 1 shows the top results by each metric for each test shape. This table includes all rows with at least one top result for any metric for the specified test shape. Moreover, we highlight the best results over all test shapes for each metric.

We see that our GAN-based approach outperforms other pipelines 2-6 times according to MTPH difference metrics: Cosine distance, Linear distance, Manhattan distance and Nuclear norm. At the same time, other methods surpass GAN in KL-divergence very insignificantly (the difference is seen only in the second or third decimal place), so this metric is almost equal.

Three of the six best GANs have been trained on a sphere shape. It has a consistent surface and consistent curvature, so this could provide better results. Moreover, the GAN tested on the sphere shows the best result on three out of five metrics, despite the large number of polygons on the sphere's surface.

We see that all pipelines tested on the key shape show better KL-divergence values than on other shapes. The key is the only mesh with a significant portion of flat elements. It could be easier to reproduce the offset distribution on flat surfaces.

You can see the examples of noise generated by our GAN in Appendix 4. It is compared with real NeRF noise.

Table 1: Top training results. The best results for a specific test shape are highlighted in green. The best metrics for all shapes are highlighted with dark green. The GAN results for the KL div. are slightly lower, however they are comparable to the rest of the approaches. The GAN results for other metrics are significantly better than others.

| Pipeline | Train on | Test on | Metrics | | | | |
|---|---|---|---|---|---|---|---|
| | | | KL div. ↓ | Cosine ↓ | Linear ↓ | Manh. ↓ | Nuclear ↓ |
| GAN | Sphere | Bird | 0.45269 | **0.00251** | **0.10487** | **2.84228** | **0.32697** |
| | All | Bottle | 0.49770 | **0.00951** | **0.21888** | **5.25660** | **0.68357** |
| | Key | Key | 0.06589 | **0.01937** | **0.32343** | **5.68141** | **1.17867** |
| | Bird | Sphere | 0.35944 | **0.00183** | **0.09029** | **2.37248** | **0.32119** |
| | Sphere | Doll | 0.53638 | **0.00625** | **0.16130** | **4.68518** | **0.46974** |
| | Sphere | Spiral | 0.49742 | **0.00368** | **0.12526** | **3.60572** | **0.31796** |
| KNN Reg. | Bird | Bird | **0.44349** | 0.01495 | 0.24551 | 6.37309 | 0.59441 |
| U-Net | All | Bottle | **0.48909** | 0.05028 | 0.56939 | 10.86997 | 1.68377 |
| DPSR + MC | – | Key | **0.06330** | 0.06551 | 0.63861 | 13.24510 | 2.26182 |
| | – | Sphere | **0.35360** | 0.13064 | 0.69201 | 20.95043 | 1.80963 |
| | – | Doll | **0.52423** | 0.09685 | 0.62921 | 19.19287 | 1.61193 |
| KNN Reg. | Spiral | Spiral | **0.48815** | 0.00552 | 0.14572 | 3.93982 | 0.44374 |

## 7 OUR DATASET EVALUATION

In the previous section we show that our pipeline can add a realistic NeRF-like noise on the mesh surface. Here we show that our pipeline can be used to produce a dataset which can upgrade learning-based denoising models. We demonstrate that learning-based denoising models can more effectively remove NeRF noise when trained on our dataset.

### 7.1 EVALUATION DATASET PREPARATION

The denoising models we have tested can only be trained on noisy-GT pairs with the same number of facets, so we prepare an evaluation dataset using the pipeline described below. The GT mesh is prepared by transformation of original mesh via DPSR + MC, as described in Section 6.

The GAN model produces a point cloud, which we label as $P$. This point cloud should be converted to a mesh with the same number of vertices as in the GT mesh. The conversion process is depicted in the Figure 7. We calculate a normal vector $n(v)$ for each vertex $v$ in GT mesh. For each $v$ we find

a point $p \in P$ the closest to the line $L_v$ defined by vertex $v$ and its normal $n(v)$. We find the point $v'$ that is closest to $v$ from line $L_v$. The $v'$ is supposed to be a point from a noisy mesh corresponding to $v$ point from the original mesh.

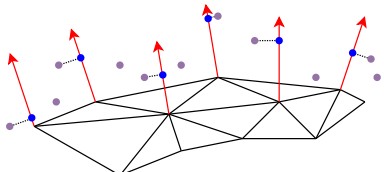 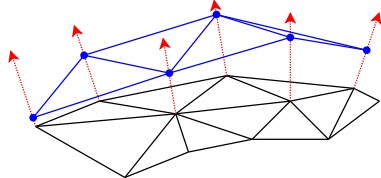

(a) The closest points to normals (red vectors) of original mesh vertices are found. These points are drawn with blue.

(b) The closest points are connected with the same edges as original mesh vertices.

Figure 7: Evaluation dataset preparation. The original mesh transformed via DPSR + MC is black. The point cloud produced by GAN is purple.

## 7.2 EVALUATION RESULTS

The GeoBi-GNN and Cascaded Regression are tested on our dataset. We train both models on six types of datasets: Synthetic only, Synthetic + GAN, Synthetic + Noisemaker3D, GAN + Noisemaker3D, GAN only and KNN-Regression only. Noisemaker3D (NM3D) is the library with a set of methods for generating node and topology noise. In addition, we prepare a dataset to be denoised by these models and measured the denoising metrics. The metrics we use for comparison are: Chamfer Distance (CD), Mean Cosine Distance of Normals (NCD), Absolute Area Difference (ADA), Mean Squared Error (MSE), and Hausdorff Distance (HD). The results are shown in the Table 2.

Table 2: Denoising metrics of GeoBi-GNN and Cascaded Regression trained with a dataset generated by our GAN and KNN-regression models. All denoising experiments are performed on NeRF-like noised meshes. The best metrics for all shapes are highlighted with dark green. The first, second and third best results shown by each model are labeled by dark green, green and light green respectively.

| Model | Train on | Metrics | | | | |
|---|---|---|---|---|---|---|
| | | CD$\downarrow$ $\times 10^{-6}$ | NCD$\downarrow$ $\times 10^{-2}$ | ADA$\downarrow$ $\times 10^{-2}$ | MSE$\downarrow$ $\times 10^{-6}$ | HD$\downarrow$ $\times 10^{-2}$ |
| GeoBi-GNN | Synthetic | 6.65 | 1.322 | 0.972 | 2.978 | 1.304 |
| | Synthetic + GAN | 8.72 | 1.141 | 1.263 | 3.899 | 0.920 |
| | Synthetic + NM3D | 6.73 | 1.217 | 0.991 | 3.019 | 1.320 |
| | GAN + NM3D | 7.86 | 1.534 | 1.233 | 3.546 | 0.906 |
| | GAN | 10.95 | 1.164 | 1.637 | 4.978 | 0.938 |
| | KNN-Regression | 6.58 | 1.753 | 1.228 | 3.142 | 0.921 |
| Cascaded Regression | Synthetic | 6.97 | 2.237 | 1.296 | 3.180 | 1.708 |
| | Synthetic + GAN | 6.77 | 2.126 | 1.198 | 3.051 | 1.683 |
| | Synthetic + NM3D | 6.96 | 2.129 | 1.295 | 3.167 | 1.734 |
| | GAN + NM3D | 6.82 | 2.172 | 1.226 | 3.073 | 1.572 |
| | GAN | 6.36 | 1.994 | 0.911 | 2.807 | 1.027 |
| | KNN-Regression | 6.42 | 1.996 | 0.896 | 2.844 | 0.980 |

The Cascaded Regression shows better results on the most of metrics being trained on our GAN-based dataset or infused with it. In particular it outperforms all other training datasets in denoising task on CD, NCD and MSE metrics. The ADA and HD metrics are comparable to KNN-based dataset. The denoising results are illustrated in Appendix 5.

### 7.3 LIMITATIONS

Besides a generative model the dataset generation pipeline includes the DPSR + MC part. Preparing the dataset for the noise generation model involves solving optimization problems, which requires a lot of time for complicated shapes with large amounts of tiny elements.

Due to the benchmark denoising models, it is necessary to use a dataset with the same number of vertices. For this reason, the preparation of the evaluation dataset requires a special transformation procedure that converts the point cloud into a mesh with the same number of vertices as the original clear mesh. This procedure is described in 7.1. This process can result in artifacts on the surface with tiny parts like birds.

The NeRF noise is determined not only by the shape topology but also by the shape texture. It is necessary to select shapes with textures that will not cause abnormal convex or concave bumps on the mesh surface after NeRF application.

## 8 CONCLUSION

In this article, we present a NeRF-like noise generation pipeline based on GAN and includes graph convolutional blocks to address challenges faced by providing reliable NeRF datasets for denoising tasks. Experimental results prove the better performance of using generated dataset for mesh denoising tasks over existing synthetic datasets. We have shown that datasets generated with our pipeline improve learning-based denoising models when used for training. The most significant improvement show the mean cosine distance and the absolute area difference of the metric normals.

Another important result is a new analysis of mesh noise that is suitable for complicated noise. Our analysis approach assumes a special heatmap calculation for vertex offsets, which has a meaning of transition probability matrix.

In future work, we want to investigate other types of mesh noise, including topology noise.

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

## A APPENDIX

### A.1 CALCULATE DISTANCE BETWEEN POINT AND MESH

In this section we present an algorithm for quick calculation of distance between point and triangle mesh. The three-dimensional space around a mesh is described as a Voronoi diagram constructed for different classes of geometric primitives that mesh consists of: facets, edges, and vertices.

Consider the point $Q$ and calculate the distance from $Q$ to the mesh. The algorithm consists of the following steps:

- Find a vertex of the mesh $A$ closest to the point $Q$. This can be done, for example, using a kd-tree calculated previously for all vertices of the mesh. We denote by $V_1, \ldots, V_n$ the vertices that are adjacent to vertex $A$. We also denote by $C_1, \ldots, C_n$ the centroids of facets adjacent to the vertex $A$.
- Denote vector $\overline{AQ}$ by $\bar{a}$. Next vectors $\overline{AV}_1, \ldots, \overline{AV}_n$ we denote by $\bar{v}_1, \ldots, \bar{v}_n$. Finally we denote vectors $\overline{AC}_1, \ldots, \overline{AC}_n$ by $\bar{c}_1, \ldots, \bar{c}_n$.
- First we should check if point $Q$ is in the reference cone of $A$.

The article clearly describes that the three-dimensional space above/below a mesh can be described as a Voronoi diagram constructed for different classes of geometric primitives. The classical Voronoi diagram is a partition of space into regions, where each region of it forms a set of points closer to one of the elements of a certain set than to any other element of the set. A mesh consists of three types of geometric primitives: facet, edge, and vertex.

The space in which the mesh is represented is transformed into a Voronoi diagram for the facets, edges, and vertices of the mesh. Drawing from the article:

In the figure, red indicates the areas where the points are closest to one of the facets than to any other facet or any of the edges or vertices. Similarly, blue indicates the areas where the points are closest to some edge, and yellow indicates some vertex.

If you want to find the shortest distance from an arbitrarily taken point to the mesh surface, then you need to take into account this feature of dividing the space around the mesh, since the distance from a point to a flat triangle in 3D is not calculated in the same way as the distance from a point to a segment or from a point to a point. It is important to understand which of the geometric primitives is closest to the point before calculating the distance.

The algorithm for finding the shortest distance can be implemented without constructing a Voronoi diagram, but with the assumption that the surface to which the distance needs to be calculated is sufficiently convex.

Suppose you want to calculate the distance from the point $Q$ to the mesh. The algorithm consists of the following steps:

1. Search for the vertex of the mesh $A$ closest to the point $Q$. This can be done, for example, using a kd-tree calculated previously for all vertices of the mesh. Denote by $V_1, \ldots, V_n$ the vertices that are adjacent to vertex $A$. We also denote by $C_1, \ldots, C_n$ the centroids of facets adjacent to the vertex $A$; 2. Check whether the point $Q$ lies in the reference cone of this vertex (in the figure these cones are indicated in yellow). To do this, take the vector connecting vertex $A$ and point $Q$, that is, vector $AQ$. Next, you need to calculate the scalar products of the vector $AQ$ with the vectors $AC_1, \ldots, AS_p$. If all these scalar products are strictly less than zero, then the point $Q$ belongs to the support cone. In this case, the desired distance is the length of the vector $AQ$. If at least one of the scalar products is greater than or equal to zero, then the distance is calculated according to the algorithm in paragraph 3; 3. For each facet $k$ adjacent to vertex $A$, calculate the vectors $L_1 C_k, L_2 C_k, L_3 C_k$, where $L_1$, $L_2$, $L_3$ are the midpoints of the facet edges. We also calculate the vectors $L_1 Q, L_2 Q, L_3 Q$, then calculate the scalar products $(L_i C_k, L_i Q), i = 1, 2, 3$. If all three scalar products are greater than or equal to zero, then the minimum distance from the point $Q$ to the mesh is equal to the distance to the facet $k$. If otherwise, the distance is calculated according to the algorithm in paragraph 4; 4. Calculate the scalar product of the vector $(AQ, AV_k), k = 1, \ldots, n$. Important: each of the vectors $V_k$ must be normalized before calculating the scalar products. Let's define k for which the scalar product $(AQ, AV_k)$ is maximal. An edge with index $k$ is the nearest edge to the point $Q$. In this case, the minimum distance from the point $Q$ to the mesh is equal to the distance to the edge $k$.

## A.2 OBJAVERSE-XL SHAPES HASHES

Table 3: Each objaverse-XL shape has a unique hash that identifies it in this dataset.

| Name | Train or test | Objaverse ID |
|------|---------------|--------------|
| bottle | Train | 00b2c8c60d2f45a893ee73fd1f107e27 |
| bird | Train | 02c81d18c4f04b9b9275fde41d0e715b |
| sphere | Train | f8c97f11180440ccae5bc156ef087014 |
| key | Train | 4bdab6b1e3194045ab6362e4c6cda222 |
| doll | Test | 0e30fca3637e4083863e1240d6d1f1bf |
| spiral | Test | 1d6ad3e20daa4873a3b1a0ab6c0ea8d1 |

## A.3 FULL RESULTS

Table 4: Basic models results: DPSR + MC, KNN, MLP, U-Net. Experiments show the best results in KL div. for a specific test shape and all shapes highlighted in green and dark green, respectively. Our GAN-based approach performs significantly better for the rest of the metrics which are shown in Table 5.

| | Test shape | Metrics | | | | |
|---|------------|---------|--------|--------|--------|--------|
| | | KL div. ↓ | Cosine ↓ | Linear ↓ | Manh. ↓ | Nuclear ↓ |
| DPSR + MC | Bird | 0.44385 | 0.02214 | 0.29057 | 8.13646 | 0.79620 |
| | Bottle | 0.49276 | 0.05177 | 0.57202 | 9.93406 | 1.60127 |
| | Key | **0.06330** | 0.06551 | 0.63861 | 13.24510 | 2.26182 |
| | Sphere | **0.35360** | 0.13064 | 0.69201 | 20.95043 | 1.80963 |
| | Doll | **0.52423** | 0.09685 | 0.62921 | 19.19287 | 1.61193 |
| | Spiral | 0.49276 | 0.07408 | 0.52683 | 15.96581 | 1.38656 |
| KNN-regressor | Bird | **0.44349** | 0.01495 | 0.24551 | 6.37309 | 0.59441 |
| | Bottle | 0.48912 | 0.03381 | 0.44590 | 9.42089 | 1.32781 |
| | Key | 0.06630 | 0.04790 | 0.48711 | 11.63991 | 1.53463 |
| | Sphere | 0.35994 | 0.08495 | 0.56878 | 16.45870 | 1.43616 |
| | Doll | 0.52934 | 0.03354 | 0.38112 | 10.68838 | 1.02367 |
| | Spiral | **0.48815** | 0.00552 | 0.14572 | 3.93982 | 0.44374 |
| MLP | Bird | 0.45572 | 0.05788 | 0.59004 | 14.28690 | 2.03749 |
| | Bottle | 0.50096 | 0.16061 | 1.43471 | 30.53953 | 5.69239 |
| | Key | 0.06677 | 0.04867 | 0.61003 | 12.47303 | 2.51558 |
| | Sphere | 0.36640 | 0.11959 | 1.04522 | 25.71050 | 3.72137 |
| | Doll | 0.54231 | 0.16903 | 0.80894 | 23.71446 | 3.58923 |
| | Spiral | 0.50105 | 0.06661 | 0.61870 | 14.49724 | 2.14522 |
| U-Net | Bird | 0.45791 | 0.12729 | 0.94130 | 18.56488 | 3.52717 |
| | Bottle | **0.48909** | 0.05028 | 0.56939 | 10.86997 | 1.68377 |
| | Key | 0.07039 | 0.27131 | 2.28507 | 40.40780 | 11.84921 |
| | Sphere | 0.36845 | 0.08761 | 0.73039 | 14.87120 | 2.56812 |
| | Doll | 0.54249 | 0.13434 | 1.06966 | 22.07343 | 4.16051 |
| | Spiral | 0.50067 | 0.13567 | 1.06968 | 24.73850 | 4.13640 |

Table 5: GAN training results. Five train datasets: *bird*, *bottle*, *key*, *sphere*, *all*. Two test datasets: *doll*, *spiral*. The best results for a specific test shape are highlighted in green. The best metrics for all shapes are highlighted with dark green. The GAN results for the KL div. are slightly lower, however they are comparable to the rest of the approaches. The GAN results for other metrics are significantly better than others.

| | Train shape | Test shape | KL div. ↓ | Cosine ↓ | Linear ↓ | Manh. ↓ | Nuclear ↓ |
|---|---|---|---|---|---|---|---|
| In domain | Bird | Bird | 0.45106 | 0.00597 | 0.18472 | 5.28831 | 0.60968 |
| | Bottle | Bottle | 0.49516 | 0.01402 | 0.26539 | 5.96482 | 0.79997 |
| | Key | Key | 0.06589 | **0.01937** | **0.32343** | **5.68141** | **1.17867** |
| | Sphere | Sphere | 0.35953 | 0.00557 | 0.15620 | 3.67948 | 0.47831 |
| Out of domain | Bottle | Bird | 0.45183 | 0.00671 | 0.17695 | 4.63796 | 0.55543 |
| | Key | Bird | 0.45828 | 0.02109 | 0.32092 | 7.61674 | 1.07289 |
| | Sphere | Bird | 0.45269 | **0.00251** | **0.10487** | **2.84228** | **0.32697** |
| | All | Bird | 0.45327 | 0.00601 | 0.16738 | 4.50107 | 0.53778 |
| | Bird | Bottle | 0.49573 | 0.01846 | 0.32097 | 6.18145 | 1.00624 |
| | Key | Bottle | 0.50158 | 0.03387 | 0.43412 | 8.31266 | 1.36748 |
| | Sphere | Bottle | 0.49697 | 0.01151 | 0.24423 | 5.57860 | 0.79836 |
| | All | Bottle | 0.49770 | **0.00951** | **0.21888** | **5.25660** | **0.68357** |
| | Bird | Key | 0.06479 | 0.02679 | 0.37076 | 6.79448 | 1.27395 |
| | Bottle | Key | 0.06527 | 0.06527 | 0.34457 | 6.19686 | 1.22980 |
| | Sphere | Key | 0.06415 | 0.03881 | 0.44968 | 8.66565 | 1.49396 |
| | All | Key | 0.06473 | 0.03096 | 0.40012 | 7.69379 | 1.39265 |
| | Bird | Sphere | 0.35944 | **0.00183** | **0.09029** | **2.37248** | **0.32119** |
| | Bottle | Sphere | 0.36089 | 0.00613 | 0.17313 | 4.32524 | 0.51328 |
| | Key | Sphere | 0.36573 | 0.01888 | 0.32437 | 8.69921 | 0.97167 |
| | All | Sphere | 0.36108 | 0.00282 | 0.10764 | 2.65000 | 0.36230 |
| Test domain | Bird | Doll | 0.53548 | 0.01027 | 0.22929 | 5.39447 | 0.67479 |
| | Bottle | Doll | 0.53560 | 0.01730 | 0.28424 | 7.04564 | 0.79888 |
| | Key | Doll | 0.54289 | 0.02074 | 0.32903 | 7.94347 | 1.14373 |
| | Sphere | Doll | 0.53638 | **0.00625** | **0.16130** | **4.68518** | **0.46974** |
| | All | Doll | 0.53805 | 0.00936 | 0.19859 | 5.59246 | 0.55628 |
| | Bird | Spiral | 0.49832 | 0.00642 | 0.18521 | 5.28779 | 0.51220 |
| | Bottle | Spiral | 0.49747 | 0.01647 | 0.29413 | 7.96902 | 0.83973 |
| | Key | Spiral | 0.50192 | 0.03417 | 0.44514 | 11.57165 | 1.42135 |
| | Sphere | Spiral | 0.49742 | **0.00368** | **0.12526** | **3.60572** | **0.31796** |
| | All | Spiral | 0.49609 | 0.00976 | 0.21889 | 6.00135 | 0.60933 |

## A.4 NeRF-LIKE NOISE EXAMPLES

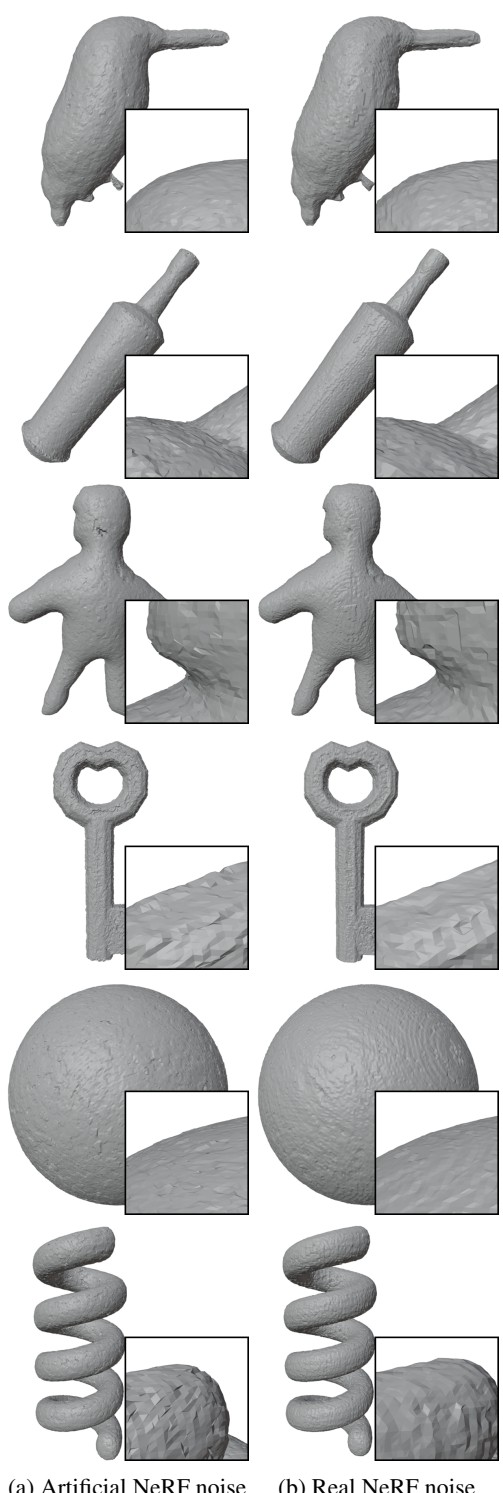

(a) Artificial NeRF noise    (b) Real NeRF noise

Figure 8: The real NeRF noise is compared to artificial noise generated by our pipeline.

## A.5 DENOISING ILLUSTRATION

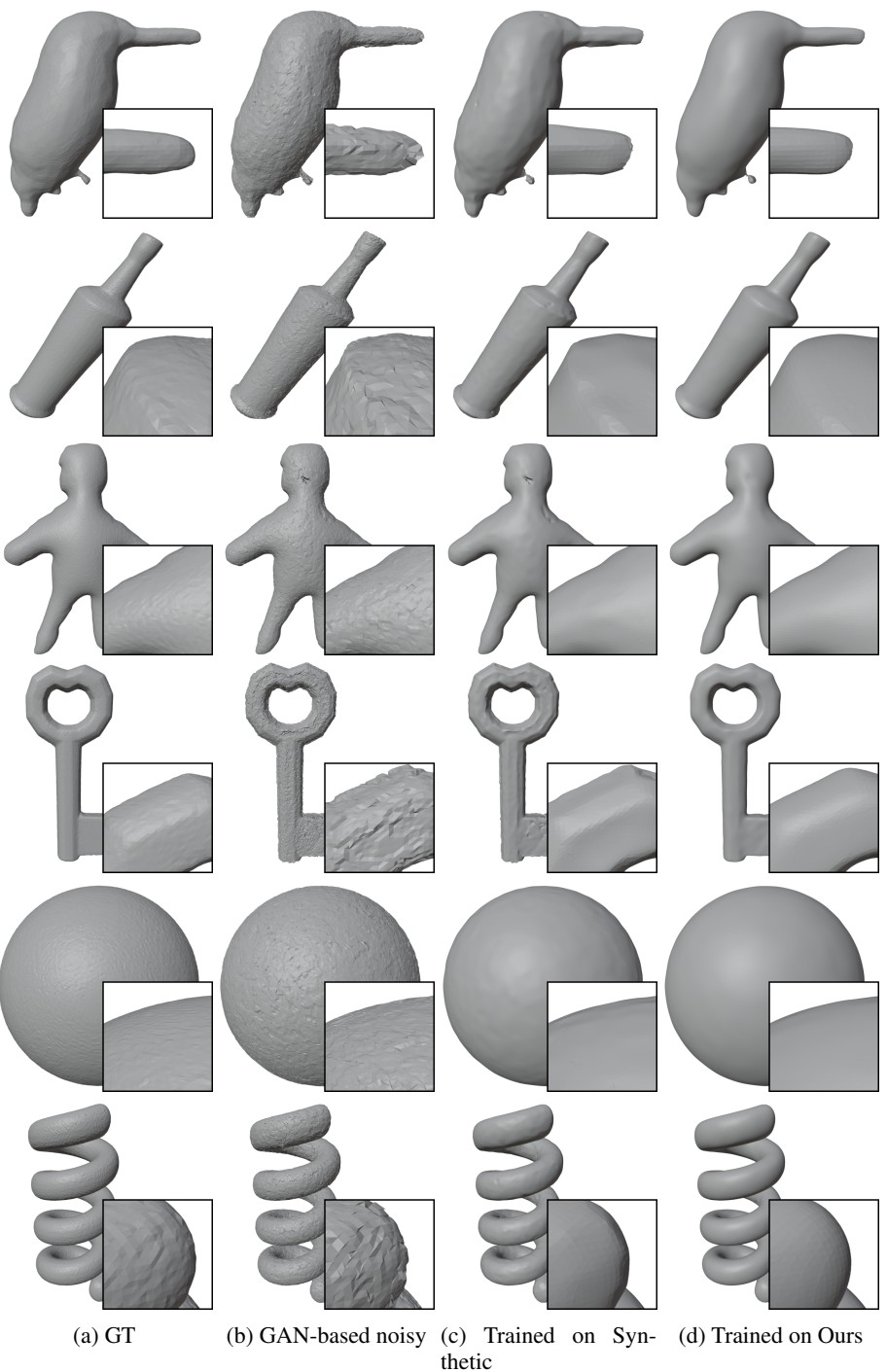

(a) GT    (b) GAN-based noisy    (c) Trained on Synthetic    (d) Trained on Ours

Figure 9: GT and noisy meshes are prepared for denoising tests as explained in Section 7.1. The denoising was performed by the Cascaded Regression model which was trained on the dataset produced by our GAN-based pipeline. We have trained Cascaded Regression on the dataset produced by KNN-based pipeline for comparison to our method. It can be seen that Cascaded Regression trained on GAN-based dataset performs better.

