# OpenReview forum: "GAN-based NeRF Noise Simulation in Mesh Denoising Task"
_ICLR.cc/2025/Conference — Submitted to ICLR 2025_

### Official Review · Reviewer_F62R · 2024-10-24

**Soundness:** 2
**Presentation:** 1
**Contribution:** 1
**Rating:** 6
**Confidence:** 3

**Summary:**

This paper proposes a network for generating Nerf like noise on mesh based on the GAN network. The corresponding dataset is also provided. This is achieved by using GAN to learn the difference between the mesh obtained by Nerf and the original one.

**Strengths:**

The GAN network is used to learn the distribution of Nerf noise, and the corresponding dataset is designed.

Demonstrating the clear difference between the Nerf noise and random noise.

**Weaknesses:**

The writing of this paper needs to be further improved, and the contribution of the work should be emphasized. It is difficult for me to find it. Is it denoising? But it seems that there are not many denoising experiments and comparison results. I would like to see a clear summary of contributions at the end of the introduction or in the abstract.

The idea of ​​this paper is simple, the method is easy to think of, and there is not much practical application shown. For example, add more applications and comparisons of mesh denoising or even point cloud denoising.

There seems to be something wrong with the reference on line 299. Should the reference format start from bpy instead of the community?

The two noise distributions shown in Figure 1 are interesting, but the mesh denoising process doesn't only consider random noise? There are also many types of noise such as white noise, blue noise, simulated scanning noise, etc. I think comparing more noise types can better highlight the special features of Nerf.

**Questions:**

Overall, this paper seems incomplete to me, with no concrete applications or innovations. And there is almost no visualization of the results. At least there should be a comparison of the errors before and after adding or removing noise to show the ability of network? And it's hard to tell why using Nerf noise. It would also be better to add more metrics for comparison after denoising/noising, such as chamfer distance, Hausdorff distance, etc.

---

### Official Review · Reviewer_GS2u · 2024-10-29

**Soundness:** 1
**Presentation:** 1
**Contribution:** 1
**Rating:** 1
**Confidence:** 3

**Summary:**

The paper proposes a GAN model to generate noisy meshes subject to a specific noise observed on meshes obtained from a NeRF 3D reconstruction.
This GAN model is then used to generate a training set to train mesh denoising algorithms on similar data.

**Strengths:**

Deals with the design of an original noise modeling.

**Weaknesses:**

Methodology:
The objective are not very clear, the processing pipeline is not clear.
NeRF do not produce meshes.
The GAN produces point cloud and not meshes.

Dataset preparation is not clear with point cloud to mesh conversion.

GeoBi-GNN method not described, no reference given for 505 "Cascaded Regression".

While the motivation for this work may be of interest, in my opinion this paper was not ready for submission.

Form:
The paper is badly written. English is poor, this looks like a draft of a submission.
Intro has too many paragraphs.
Figures are badly presented, Appendix is not provided at the end of the paper.
p1 SDF: meaning?
p1: contributions using past tense?
Some passages look like a student report with a lot of details that are not relevant:
 - NeRF space in intro
 - p. 3 architectures
 - p. 3 line 135
 - p. 4 GAN
 - l. 335 min (Rigid point-set registration is a standard technique)
 No green in table 2 and 3 shades of green in Table 1.

**Questions:**

Can a GAN produce directly a noisy mesh ?
Why extract meshes from NerF?
Isn't there better procedure than Marching Cube for that?
Isn't there NeRF approaches based on meshes?

---

### Official Review · Reviewer_LP8m · 2024-11-03

**Soundness:** 3
**Presentation:** 3
**Contribution:** 3
**Rating:** 6
**Confidence:** 4

**Summary:**

This paper presents an intriguing approach to generating NeRF-like noise on mesh surfaces, along with a dataset tailored for this purpose. The authors construct a dataset and train a GAN for NeRF-based mesh denoising. I appreciate the innovative Mesh Transition Probability Heatmap (MTPH) concept, which accounts for the influence of a vertex’s neighborhood rather than focusing on individual vertices alone. This neighborhood-aware approach leads to a more accurate analysis of noise distribution.

However, I believe the authors could improve the clarity of their presentation. For example, Figure 1 effectively contrasts random noise with NeRF noise, and I can clearly observe the structured pattern characteristic of NeRF noise. However, the figure lacks crucial contextual details—such as the structure of the mesh and the meaning of the axes—that would aid in interpreting the depicted patterns. This missing information makes Figure 1 somewhat confusing.

Additionally, while the authors report quantitative metrics to evaluate their results, I would like to see more visual comparisons, such as rendered images, that illustrate the proposed denoising approach’s effectiveness. Unfortunately, such visual comparisons are absent, which could have provided valuable insight into the visual improvements achieved by the method.

**Strengths:**

This paper constructs a dataset consisting of pairs of original and NeRF-noised meshes, creating a valuable resource for future denoising applications. Additionally, it proposes a novel approach—Mesh Transition Probability Heatmap (MTPH)—to measure noise by considering both vertex offsets and their neighboring relationships. The authors effectively demonstrate the strength of this method through superior performance results. The paper also presents a thoughtfully designed pipeline for dataset generation, training, and evaluation, underscoring the importance and success of their algorithm.

**Weaknesses:**

I appreciate the thorough evaluation conducted by the authors, but I believe it is essential to include denoised renderings to visually highlight the differences between their approach and others. Additionally, using the DPSR + MC method for dataset preparation may impose constraints, particularly for meshes with complex topologies, and this limitation should be discussed in the paper. The only visual example provided is of a bird model, which may not be sufficient to demonstrate the approach’s advantages comprehensively. I strongly recommend including more visual examples to enhance the clarity of the results.

Moreover, while the dataset and model show promising results within a controlled setting, further validation on a broader range of real-world meshes would strengthen claims about the approach’s practical applicability.

**Questions:**

1. How should I interpret Figure 1? Please add more annotations and explanations in both the figure and its caption to clarify its components and meaning.
2. Since MTPH is calculated based on each vertex and its neighbors, it inherently depends on mesh connectivity. Have the authors tested their model with the same object but with different mesh connectivities? If so, what were the findings?

---

> ### Author Response · Authors · 2024-11-21
> **Rebuttal by Authors**
>
> Thank you very much for reading our work and for your insightful suggestions!
>
> __Constraints:__
>
> Speaking about the constraints highlighted in the paper, we considered constraints that are mentioned at line 297: complicated topology (we do not consider meshes with Euler-Poincar´e characteristic greater than 10). However we did not emphasize that we filtered out the meshes with tiny elements (for example fishes with little fins) because the DPSR + MC (and  fitting mesh pairs procedure too) were not perfect for them. These limitations will be concluded in the camera-ready version.
>
> __About MTPH dependency on mesh connectivity:__
>
> We calculated the MTPH only for the meshes with one connected component. However, it is quite possible to compute it for each connected component separately. This is a statistical approach, and it requires considering the vertex neighborhood regardless how many components the mesh has.
>
> __Corrections:__
>
> --Figure 1 is intended to show that the random noise applied for each vertex separately corresponds to the MTPH which looks like a constant along the vertical axis. At the same time the NeRF noise causes complicated dependencies between vertices: each vertex has only the neighbors with offsets close to its own offset. We will make clear the Figure 1 caption  if given the opportunity.
>
> --We appreciate your recommendation to put the meshes visualization into the appendix to show how our model performs. We will add examples into the camera-ready version, should we have the opportunity.

---

> ### Author Response · Authors · 2024-12-02
>
> Dear reviewer LP8m, could you please update us on the response? Thank you again for reading our work and for your comments.

---

### Meta-Review · Area_Chair_jp8n · 2024-12-18

**Metareview:**

This paper proposes a method for generating NeRF-like noise on mesh surfaces, along with a dataset specifically designed for this task. The authors introduce a GAN-based network that generates NeRF-like noise on meshes by learning the difference between the mesh obtained via NeRF and the original mesh. A key contribution is the Mesh Transition Probability Heatmap (MTPH), which accounts for the influence of a vertex’s neighborhood, rather than focusing solely on individual vertices.

All reviewers identified two critical issues with the submission: 1) the lack of clear explanations regarding the potential applications of the proposed method, and 2) unclear and disorganized presentation. After the rebuttal, all reviewers agreed on these two points, and the discussion quickly converged on a recommendation for rejection. The AC concurs with these concerns and has also decided to reject the submission.

**Additional Comments On Reviewer Discussion:**

Please see the Metareview.

---

### Decision · Program_Chairs · 2025-01-22

Reject